# Pre-Emphasis Pulse Design for Random-Access Memory

**Yoshihiro Sugiura and Toru Tanzawa ***

Faculty of Engineering, Shizuoka University, Hamamatsu 432-8561, Japan; sugiura.yoshihiro.16@shizuoka.ac.jp
* Correspondence: toru.tanzawa@shizuoka.ac.jp

**Abstract:** This paper describes how one can reduce the memory access time with pre-emphasis (PE) pulses even in non-volatile random-access memory. Optimum PE pulse widths and resultant minimum word-line (WL) delay times are investigated as a function of column address. The impact of the process variation in the time constant of WL, the cell current, and the resistance of deciding path on optimum PE pulses are discussed. Optimum PE pulse widths and resultant minimum WL delay times are modeled with fitting curves as a function of column address of the accessed memory cell, which provides designers with the ability to set the optimum timing for WL and BL (bit-line) operations, reducing average memory access time.

**Keywords:** pre-emphasis pulse; random-access memory; RC delay; behavior model

## 1. Introduction

Nonvolatile random-access memory (NVRAM) or storage class memory are bridging the gap between volatile main memory (DRAM) and nonvolatile NAND flash memory in the memory hierarchy in terms of memory access time to improve memory performance [1,2]. In addition to much faster access time than NAND, NVRAM costs much less than DRAM, helping to keep the computer system cost effective. 3D cross-point memory structure has come to a solution to cost scaling in more advanced nonvolatile memory technology by increasing the number of nonvolatile memory layers [3–7]. A design guideline was proposed for 3D cross-point memory to have a sufficient operation margin to read and write in [8].

Pre-emphasis (PE) pulses are design techniques used to reduce access line delay, especially in large arrays such as 3D NAND [9] and large flat panel display [10]. By driving large RC delay lines with a pulse whose initial period is made with a voltage higher than the target voltage, the entire delay time can be reduced significantly, where the delay time is defined by the farthest point of the line. In [9,10], two calibration methods were proposed, since a precise PE pulse is required even with process variation in the RC time constant. In [11,12], a circuit analysis is discussed to design the PE pulse for minimizing the delay time. Based on the circuit analysis, a PE pulse generator with feedback was proposed in [13].

In this paper, PE pulse design is discussed for NVRAM, where the delay time depends on the column address. Hence, an optimum pulse width of the PE pulse can vary according to the position of the selected memory cell across a selected word-line (WL). In Section 2, the optimum PE pulse width and the minimum delay time are identified as a function of the position on WL in cases of an ideal case with no process variation and an actual case with process variation. Impact of cell current and resistance of decoding transistors is also investigated. In Section 3, the simulated data is compared with measured data for validation. In Section 4, WL behavior with PE pulses is expanded to a three-lines model. Fitting curves with a limited number of parameters are presented for the optimum PE pulse width and the minimum delay time across WL to design the pulses. Colum address dependent PE pulse width is proposed and is applied to a memory system.

## 2. Optimum Pre-Emphasis Pulse Design

Figure 1a illustrates a memory array including four different positions $N_1$–$N_4$ across WL, each of which is located at x = 1/4, 1/2, 3/4, and 1. Figure 1b shows simulated waveforms when WL is driven by a PE pulse with an emphasis $\alpha$ of 1.5. As shown in Figure 1c, when the delay time is defined with a voltage window $\beta$ of 10%, the cell at $N_2$ has the shortest delay time among the four points because the nearest cell at $N_1$ has an overshoot over 10%. Thus, there should be an optimum pulse width per position. An $\alpha$ of 1.5 and a $\beta$ of 10% are used as the nominal conditions in this paper unless otherwise specified. All the values in this paper that have second as a unit can be scaled by the time constant of WL RC. Thus, arbitrary units are used for time-related parameters.

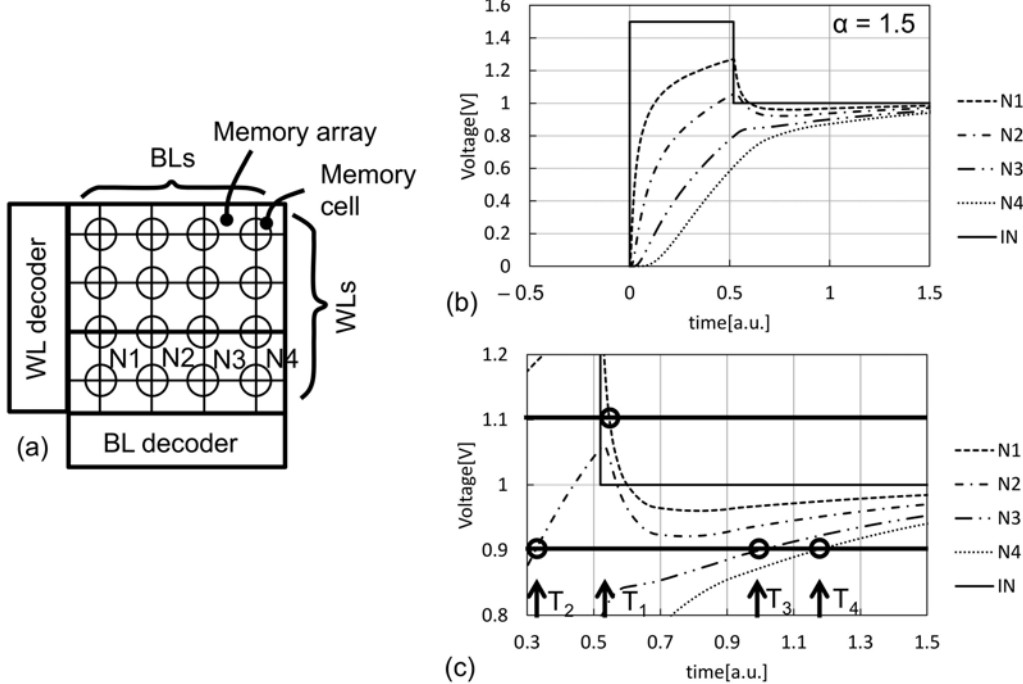

**Figure 1.** (**a**) Memory array, (**b**,**c**) WL behavior at x = 1/4, 1/2, 3/4, and 1 with PE pulse where $\alpha$ and $\beta$ are 1.5 and 0.1, respectively.

To see how the PE pulse width $T_{PRE}$ affects the WL delay time $T_{DLY}$, $T_{PRE}$ is skewed as shown in Figure 2a–d. An $\alpha$ of 1.5 and a $\beta$ of 10% are demonstrated. When $T_{PRE}$ is shorter than optimum, WL at the target $N_2$ does not reach 90% of the target voltage, resulting in a longer $T_{DLY}$ than the minimum, as shown in Figure 2a. When $T_{PRE}$ is longer than optimum, WL at the target $N_2$ overshoots, resulting in a longer $T_{DLY}$ than the minimum, as shown in Figure 2b. Figure 2c shows the minimum $T_{DLY}$. As one can imagine, there is a window in $T_{PRE}$ to have the minimum $T_{DLY}$, as shown in Figure 2d.

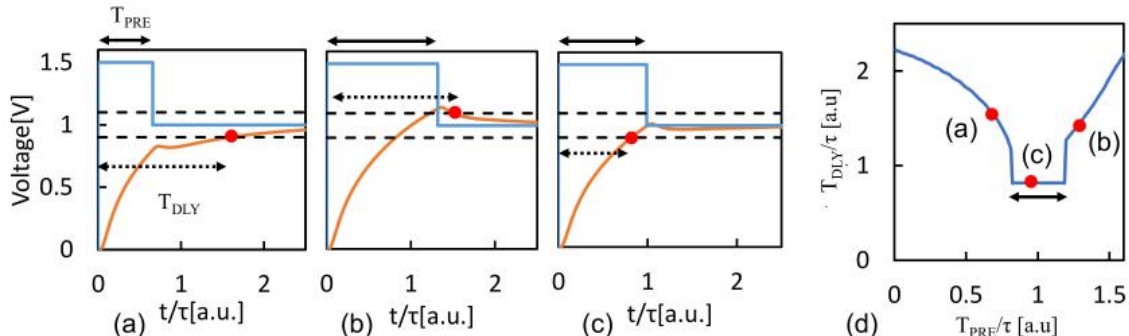

**Figure 2.** WL behavior at $N_2$ with $T_{PRE}$ of $0.6\tau$ (**a**), $1.25\tau$ (**b**), and $0.9\tau$ (**c**), and $T_{DLY}$ vs. $T_{PRE}$ (**d**).

### 2.1. Ideal Case with no Process Variation

When $T_{PRE}$ was varied, there were four patterns in WL waveform at different locations across WL, as summarized in Table 1. In the case of an $\alpha$ of 1.5 and a $\beta$ of 10%, those patterns are distributed as shown in Figure 3. The vertical axis is $T_{PRE}$ normalized by $T_{OPT}$, as given by (1), which is the optimum $T_{PRE}$ in case of NAND where the delay time is determined by the farthest location in WL. $\tau$ is a time constant given by $\tau = 4RC/\pi^2$.

$$T_{OPT} = \tau \ln \frac{\alpha}{\alpha - 1} \tag{1}$$

Two boundaries indicate $T_{DLY}$ can be minimized at the location x when $T_{PRE}$ is set between those two boundaries. As expected, the minimum can be realized with pattern 1 or 2. However, below 20% of x, there is no $T_{PRE}$ to realize pattern 1 or 2. This is because an $\alpha$ of 1.5 is too large to realize pattern 1 or 2 with a $\beta$ of 10%.

**Table 1.** WL waveform pattern.

| Pattern | What Point Determines $T_{DLY}$ | Waveform |
|---------|--------------------------------|----------|
| 1 | First rising point in higher order behavior | |
| 2 | First rising point in single order behavior | |
| 3 | First falling point in higher order behavior | |
| 4 | Second rising point in higher order behavior | |

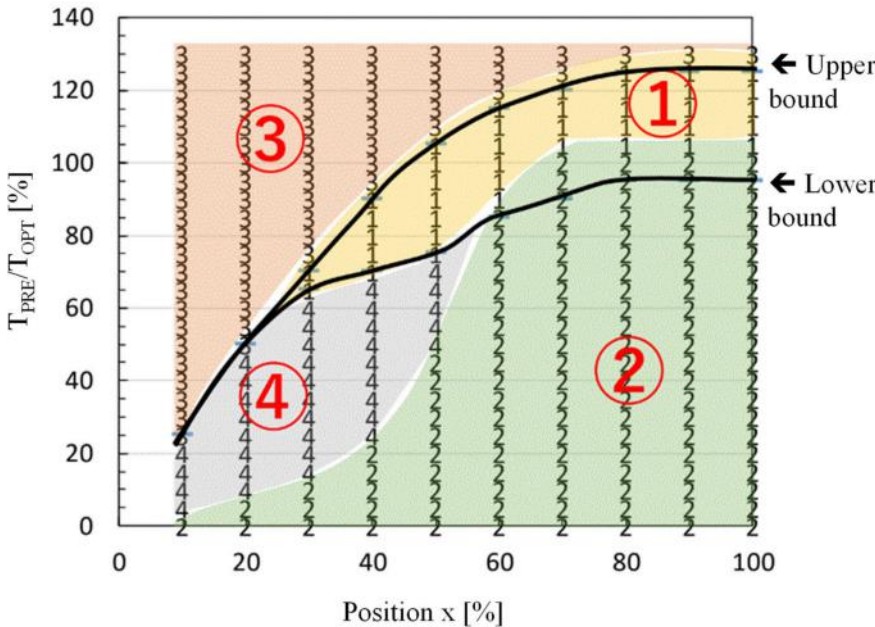

**Figure 3.** WL waveform pattern across x and $T_{PRE}$. 1–4 areas in the figure indicate pattern 1–4 in Table 1, respectively.

Figure 4 shows $T_{DLY}$ as a function of $T_{PRE}$ for x = 1/6, 1/3, 1/2, and 1. A dot indicates the optimum point for the case of NAND. As one can see, All four curves penetrate that

point, which means that all points have the same time delay with the same $T_{PRE} = T_{OPT}$. On the other hand, each point has a different optimum $T_{PRE}$ with its own minimum delay time. For example, the minimum $T_{DLY}$ of x = 1/6 is about 0.5τ, with a $T_{PRE}$ of 0.47τ, whereas the minimum $T_{DLY}$ of x = 1/2 is about 0.8τ, with a $T_{PRE}$ of 0.85τ–1.15τ.

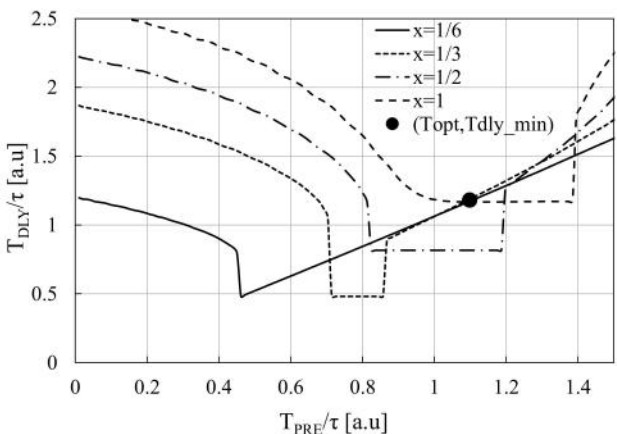

**Figure 4.** $T_{DLY}$ as a function of $T_{PRE}$ for x = 1/6, 1/3, 1/2, and 1.

Figure 5a shows which positions can have a shorter delay time when $T_{PRE}$ is set to a specific value. For x < 0.5, the range of optimum $T_{PRE}$ does not include $T_{OPT}$, resulting in a significant difference in $T_{DLY}$ between the case of $T_{PRE} = T_{OPT}$ and that of an optimum $T_{PRE}$ at each x. Bit-line (BL) delay time starts with WL high. As a result, there is room to start BL access earlier for, e.g., x < 0.8. The memory system taking advantage of that feature will be discussed later. Figure 5b compares $T_{DLY}$ with α = 1.5 and α = 1.2 when $T_{PRE}$ is determined to have the minimum $T_{DLY}$ at each x. A higher pre-emphasis pulse height significantly reduces $T_{DLY}$ at x > 1/3, but increases $T_{DLY}$ a little at x < 1/3 because a larger α does not realize the fastest pattern 1 in Table 1 at the near end of WL.

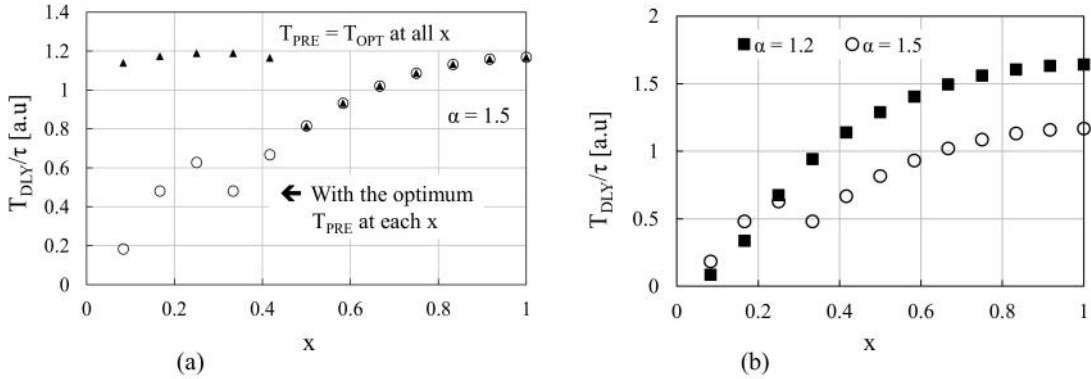

**Figure 5.** $T_{DLY}$ as a function of x. $T_{DLY}$ is compared (**a**) between the case where $T_{PRE}$ is set at $T_{OPT}$ at all x and the case where $T_{PRE}$ is determined to have the minimum $T_{DLY}$ at each x and (**b**) between α = 1.5 and α = 1.2 when $T_{PRE}$ is determined to have the minimum $T_{DLY}$ at each x.

### 2.2. Experiment

Figure 6a shows a chip micrograph to validate the SPICE simulation. The test circuits were fabricated in a 0.18 μm 3V CMOS [14]. The RC line is made of multiple units of RC elements, where R and C are given by the poly resistor and MIM capacitor, respectively. Even when a different process technology is used, all the graphs in this paper are still valid because the performance parameters such as $T_{PRE}$ and $T_{DLY}$ are normalized by the RC time constant. Internal nodes can be measured with analog buffers as shown in Figure 6b. Figure 6c is a measured waveform at x = 1/3 and 1.

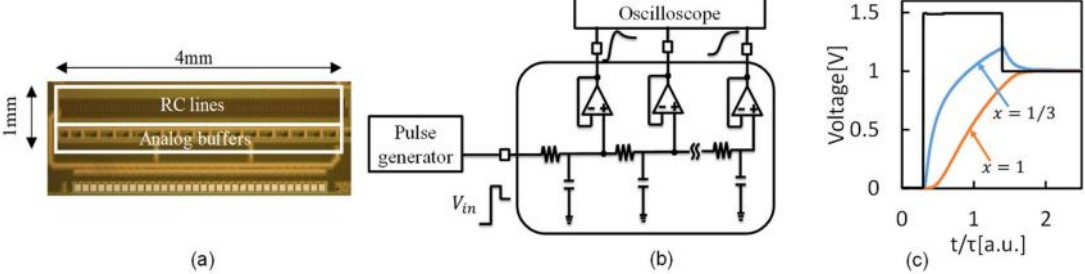

**Figure 6.** (**a**) die photo, (**b**) block diagram of the test circuit, and (**c**) voltage waveform at x = 1 and 1/3 ($\alpha$ = 1.5).

$T_{DLY}$ is measured with $T_{PRE}$ varied at x = 1/6, 1/3, 1/2, and 1 to determine optimum $T_{PRE}$ for minimizing $T_{DLY}$ at each x. Except for x = 1/6, $T_{PRE}$ has the window whose edge points are plotted in Figure 7a. With such optimum $T_{PRE}$ at each x, $T_{DLY}$ is given as a function of x, as shown in Figure 7b. Table 2 summaries errors of measurement with SPICE.

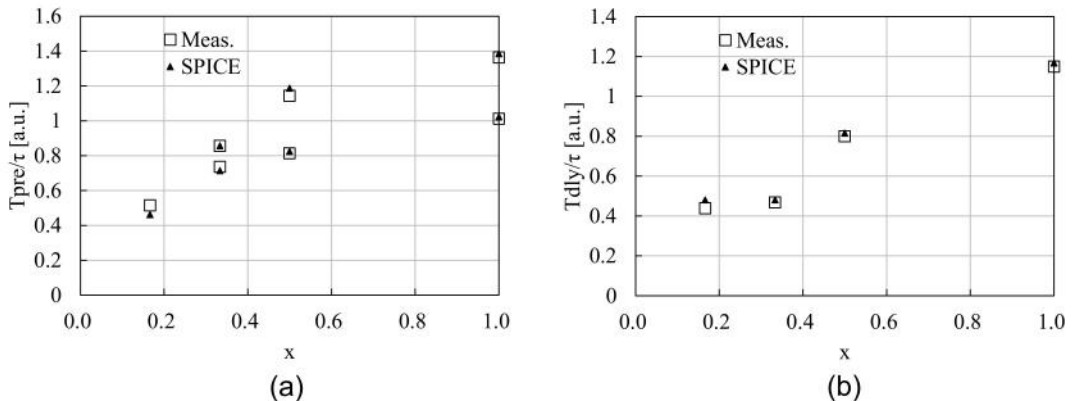

**Figure 7.** Optimum $T_{PRE}$ vs. x (**a**) and $T_{DLY}$ vs. x (**b**).

**Table 2.** Error between measured $T_{DLY}$ and simulated $T_{DLY}$ at each x.

| x | Error [%] |
|---|---|
| 1/6 | 12.2 |
| 1/3 | 6.0 |
| 1/2 | 2.0 |
| 1 | 1.5 |

*2.3. Worst Corner under Process Variation*

Figure 8 shows $T_{DLY}$ as a function of $T_{PRE}$ under the corner condition of x = 1/2, where RC varies by ±20%. The nominal corner shown by 0% is the same curve as the one in Figure 4. When RC increases, the nominal curve simply shifts in the right top direction of 45°. Therefore, the $T_{DLY} - T_{PRE}$ region can be given as in grey. As a result, the worst corner is determined by the curve in red. Below about 1 for normalized $T_{PRE}$, the corner of +20% determines $T_{DLY}$, whereas over about 1 or normalized $T_{PRE}$, the corner of −20% determines $T_{DLY}$.

Such a corner is gathered for different locations x, as shown in Figure 9. Every curve has no flat region in terms of $T_{PRE}$. The vertical line marked as "$T_{OPT}$" indicates the case when (1) is applied. The graph suggests that $T_{DLY}$ is reduced for x < 1 even with $T_{PRE}$ = $T_{OPT}$. It also suggests that $T_{DLY}$ can be minimized if one sets $T_{PRE}$ to the lowest point.

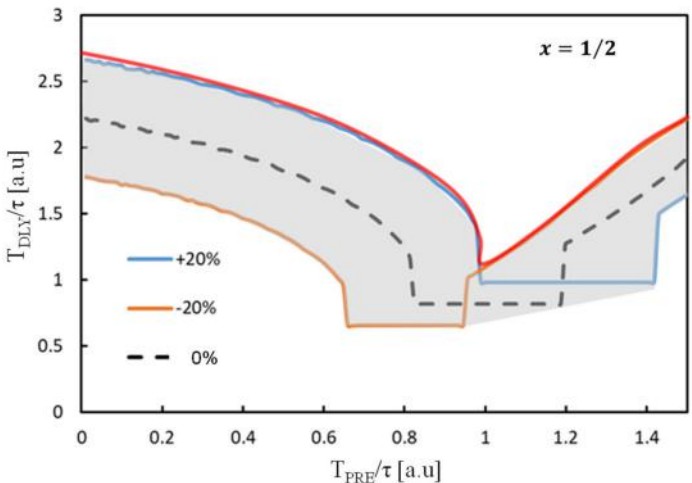

**Figure 8.** $T_{DLY}$ as a function of $T_{PRE}$ under the worst corner.

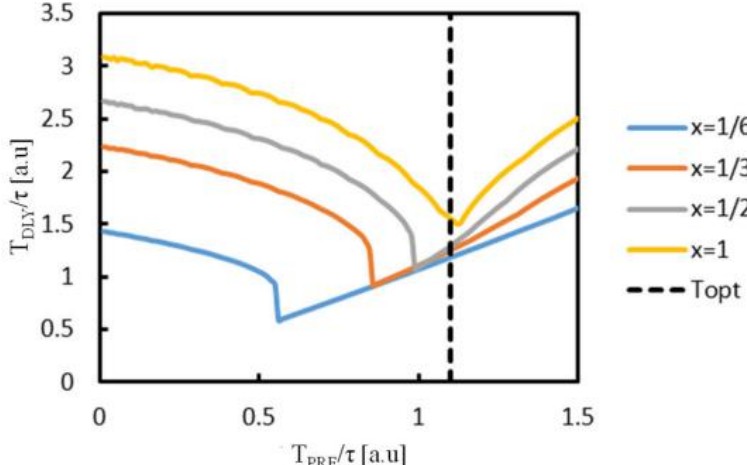

**Figure 9.** $T_{DLY}$ vs. $T_{PRE}$ at x = 1/6, 1/3, $\frac{1}{2}$, and 1 under the worst corner.

Figure 10 shows optimum $T_{PRE}$ vs. x under the nominal and worst corners based on the data of Figures 4 and 8. One can set $T_{PRE}$ with $T_{OPT}$ for x > 1/2. On the other hand, it is good to set $T_{PRE}$ with optimum values for x < 1/2.

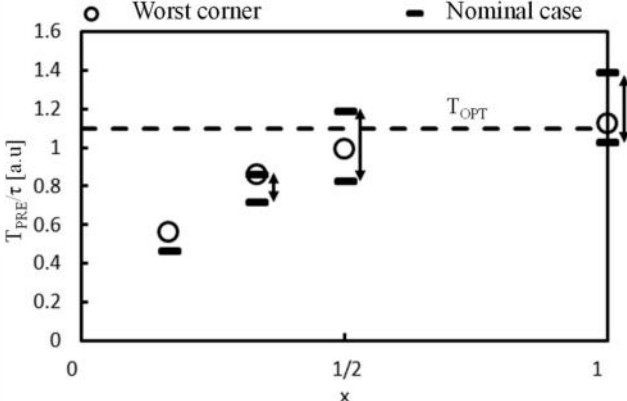

**Figure 10.** Optimum $T_{PRE}$ vs. x under the nominal and worst corners.

Figure 11 shows that x-dependent optimum $T_{OPT}$ can reduce $T_{DLY}$ by 50% at most.

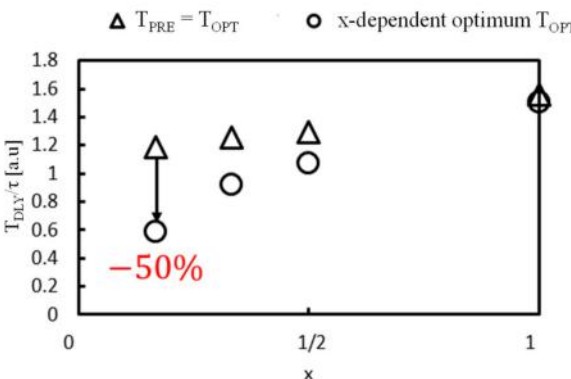

**Figure 11.** $T_{DLY}$ vs. x under the worst corner.

### 2.4. Impact of Cell Current

3D cross point memory has non-volatile memory cells, each of which flows a cell current. The cell current depends on data 0, 1. When pre-emphasis pulses are used for such memory, an impact of cell current needs to be validated. Figure 12 illustrates WL line model when cells flow at the cell current, which is modeled by Rcel. Let us introduce a parameter $\gamma$ as Rcel = $\gamma$ R. When $\beta$ = 0.1, $\gamma$ must be greater than 9. Otherwise, the WL voltage at x = 1 cannot reach 0.9E. As $\gamma$ decreases, $T_{DLY}$ should increase.

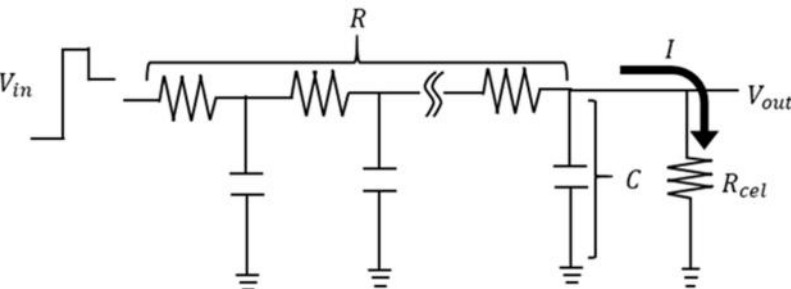

**Figure 12.** WL line model when cells flow at cell current.

As shown in Figure 13, when $\gamma$ = 10, $T_{DLY}$ increases by 1~7% across WL. However, for $\gamma$ > 30, $T_{DLY}$ only increases 1% at most. Such an analysis is needed to determine the maximum WL length. Once Rcel is determined, WL length should meet the condition R < Rcel/30.

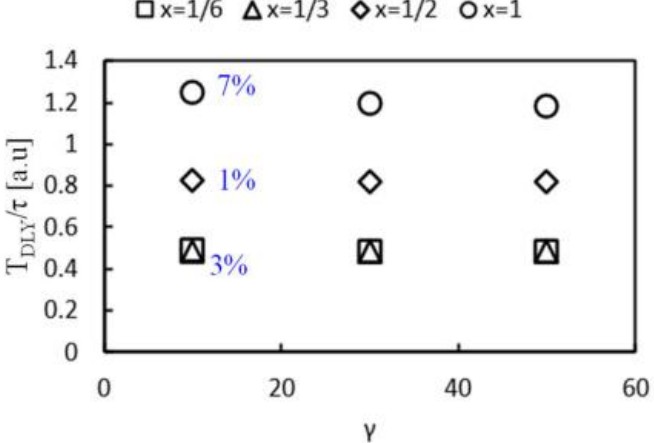

**Figure 13.** $T_{DLY}$ vs. $\gamma$ with x-dependent optimum $T_{PRE}$.

### 2.5. Impact of Decoding Transistors

Another concern when designing pre-emphasis pulses for random-access memory is the impact of the driver resistance Rd including decoding transistors and wiring resistance on optimum $T_{PRE}$ and $T_{DLY}$ (see Figure 14). Let us introduce $\delta$ to define Rd by Rd = $\delta$ R.

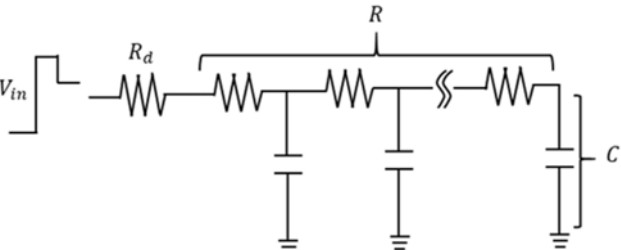

**Figure 14.** WL line model when the driver resistance is considered.

Figure 15a–d show optimum $T_{PRE}$ and $T_{DLY}$ at x = 1/6, 1/3, 1/2, and 1 as a function of $\delta$. Optimum $T_{PRE}$ increases in proportion to $\delta$ at a rate of about 2.5~3 regardless of x. $T_{DLY}$ has a similar tendency to $\delta$, except when x = 1/6. A finite $\delta$ affects $T_{DLY}$ at x = 1/6 most, which decreases $T_{DLY}$ with $\delta$ = 0.3~0.5 at most.

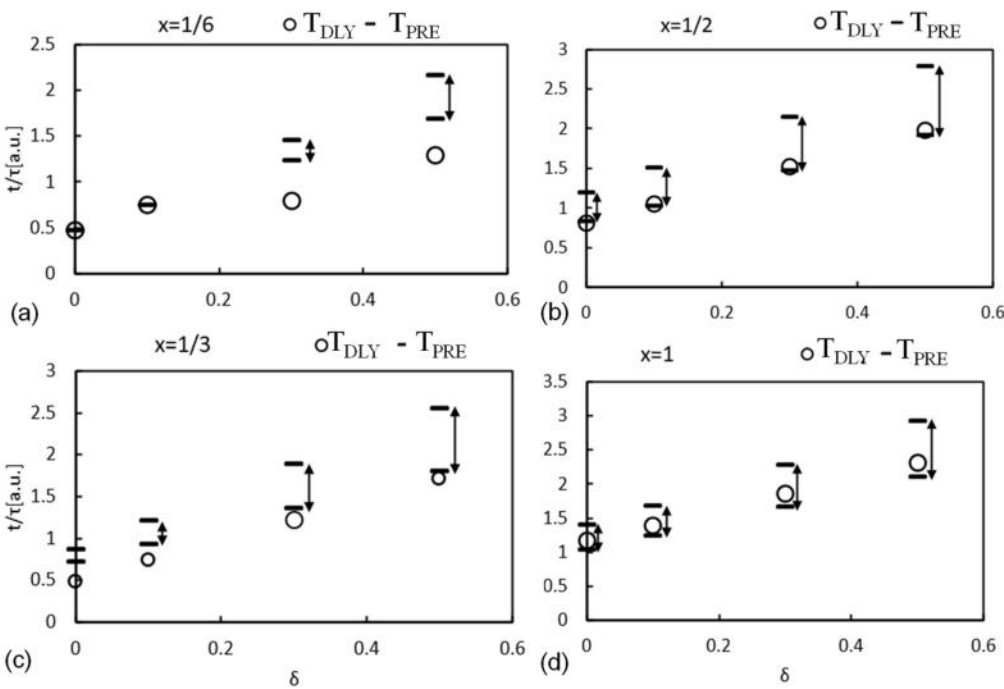

**Figure 15.** Optimum $T_{PRE}$ and $T_{DLY}$ vs. $\delta$ for x = 1/6 (**a**), 1/3 (**b**), 1/2 (**c**), and 1 (**d**).

### 3. Discussion

In this section, three-line cases, fitting curves for Optimum $T_{PRE}$ and $T_{DLY}$, and applications for memory systems are discussed.

### 3.1. Three-Line Model

In this section, the general three-line model shown in Figure 16 is studied. The center line is a target delay line, while the next neighbor lines are grounded. The lines have grounded capacitors Cg and coupling capacitors Cc.

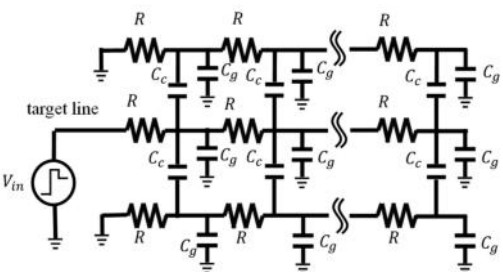

**Figure 16.** General three-line model with grounded cap Cg and coupling cap Cc.

The time constant τ of the three-line model is approximated by the sum of RCg and 3RCc;

$$\tau = R(Cg + 3Cc), \tag{2}$$

as shown in Figure 17.

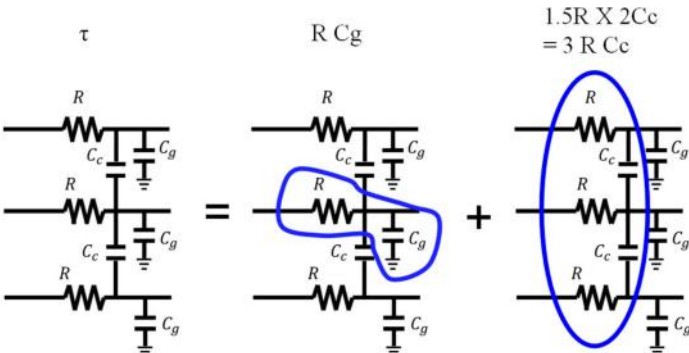

**Figure 17.** Effective time constant model.

SPICE simulations were done for the two cases where Cc:Cg = 1:1 and 100:1 under the condition that τ varies by ±20% and α = 1.2, β = 0.1, resulting in Figure 18. Even though the capacitor ratios are very different, the $T_{DLY}$ vs. $T_{PRE}$ curves normalized by each nominal τ are well matched at four different locations.

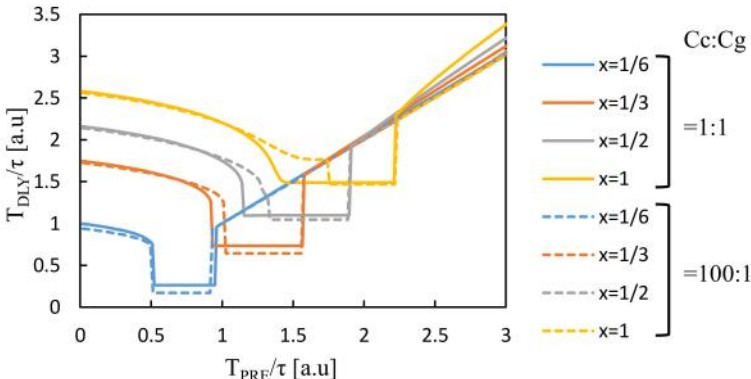

**Figure 18.** $T_{DLY}$ vs. $T_{PRE}$ for the models with different cap ratios under the worst case where τ varies by ±20% (α = 1.2, β = 0.1).

*3.2. Fitting Curve*

To see if one can fit $T_{DLY} - x$ and $T_{PRE} - x$ curves for those two conditions on the cap ratio with single equations with a few fitting parameters, the following equations were investigated.

$$T_{PRE} = \gamma_1\, T_{OPT\_NAND}\, [1 - \exp(-x/\mu_1)], \tag{3}$$

$$T_{DLY} = \gamma_2\, T_{DLY\_NAND}\, [1 - \exp(-x/\mu_2)], \tag{4}$$

where $T_{OPT\_NAND}$ and $T_{DLY\_NAND}$ are given by (5) and (6), respectively, as given for NAND [8,9].

$$T_{OPT\_NAND} = \tau \ln \frac{\alpha}{\alpha - 1} \tag{5}$$

$$T_{DLY\_NAND} = \frac{\tau}{9} \ln \left| \frac{4\alpha}{3\pi\beta} \left\{ \left( \frac{\alpha}{\alpha-1} \right)^8 - 1 \right\} \sin \frac{3\pi x}{2} \right|, \tag{6}$$

When one uses $\gamma_1 = 1.2$, $\mu_1 = 0.4$, $\gamma_2 = 0.9$, and $\mu_2 = 0.8$, the curves are well fit, as shown in Figure 19. "up" and "low" indicate the upper and lower bounds in $T_{PRE}$ to have the minimum delay time. The fitting curve for $T_{PRE}$ vs. x was well done within the upper and lower bounds of both 1:1 and 100:1. Therefore, one needs only two independent fitting parameters per specific $\alpha$ and $\beta$. The fitting curve for $T_{DLY}$ vs. x was not as well done as the one for $T_{PRE}$ vs. x, but it did validate that a moderate fitting could be done with only two fitting parameters as well. Thus, such a behavioral model allows designers to set optimum PE pulse widths and resultant delay times.

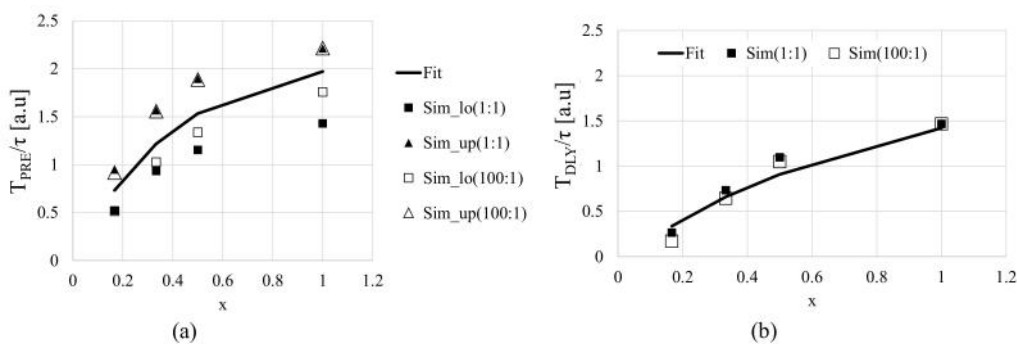

**Figure 19.** Fitting and simulated curves for $T_{PRE}$ vs. x (**a**) and $T_{DLY}$ vs. x (**b**), whose data is taken from Figure 18.

### 3.3. Application to Memory System

NVRAM is expected to have much faster access time than NAND or NAND-based solid-state drive, and to have a moderate access time and low bit cos in comparison with DRAM. In such a situation, WL and BL delay times can be as long as multiple clock periods due to large memory arrays. Column-address-dependent memory access can reduce the WL latency when the memory cells located close to the WL decoder are accessed. Figure 20 illustrates a block diagram to realized column-address-dependent memory access. The pre-emphasis pulse controller varies the PE pulse width depending on the column address. The following operations, including BL access and I/O control, can start earlier than the case where the memory cells located at the far side of WL are accessed. The memory controller and CPU can synchronize with it because they know the column address.

When the number of clocks required for WL rise is $N_{WL}$, which depends on the column address, and that for the other delay times from the address input to the WL decoder and from the memory array to the output buffer is $N_{REST}$, the total latency of the NVRAM is given by $N_{WL} + N_{REST}$. Assuming $N_{WL}$ varies from 2 at the nearest cell access to 15 at the farthest cell access based on Figure 19b, one can draw the latency improvement expressed by $1 - (N_{WL\_AVG} + N_{REST})/(N_{WL\_WORST} + N_{REST})$ as a function of $N_{REST}$ as shown in Figure 21 where $N_{WL\_WORST}$ is the worst case $N_{WL}$ when the farthest cell is accessed and $N_{WL\_AVG}$ is the average value of $N_{WL}$ between when the farthest and nearest cells are accessed. When NVRAM is designed to have an $N_{REST}$ of 5 to 20, the average latency can be improved by 20–30% with the proposed operation.

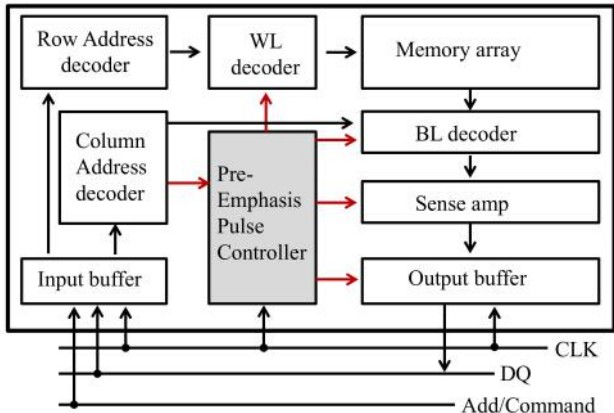

**Figure 20.** Block diagram of a proposed NVRAM.

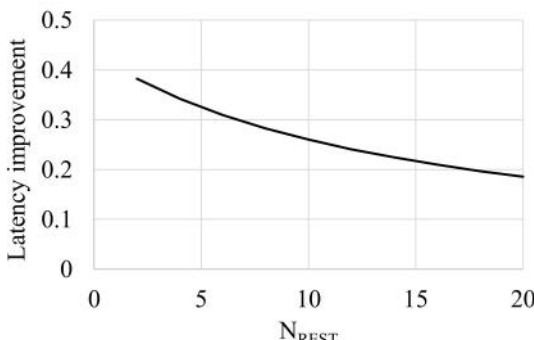

**Figure 21.** Latency improvement with the proposed NVRAM operation.

## 4. Conclusions

PE pulsing was studied to assess whether one can reduce the memory access with a PE pulse even when the memory is a random-access type. One can design a PE pulse whose width varies by column address to reduce the WL delay time, even under process variation. The impact of the cell current and the resistance in the decoding path on optimum PE pulse widths and resultant WL delay times are also investigated. Fitting the curves of optimum PE pulse widths and resultant WL delay times of as a function to the column address are demonstrated using only two parameters for each in the case of $\alpha = 1.2$, $\beta = 0.1$, and process variation in $\tau$ of $\pm 20\%$. A block diagram is also proposed to allow column-dependent memory operations to have faster average access.

**Author Contributions:** Conceptualization, T.T.; methodology, Y.S. and T.T.; software, Y.S.; validation, Y.S. and T.T.; formal analysis, Y.S. and T.T.; investigation, Y.S. and T.T.; writing—original draft preparation, T.T.; writing—review and editing, Y.S. and T.T.; funding acquisition, T.T. Both authors have read and agreed to the published version of the manuscript.

**Funding:** This research was partially funded by Kioxia Corp.

**Acknowledgments:** This work is supported by Kioxia Corp., VDEC, Synopsys, Inc., Cadence Design Systems, Inc., Rohm Corp., and the Micron Foundation.

**Conflicts of Interest:** The authors declare no conflict of interest.

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
