# Peer review of "Pre-Emphasis Pulse Design for Random-Access Memory"

_electronics, doi:10.3390/electronics10121454_

Round 1

Reviewer 1 Report

  1. An interesting paper regarding a pre-emphasis pulse design for RAM
  2. Some figures are not clear enough, ex. Figure 1(a), 1(b), 3, and 15. please update the clearer figures.
  3. Authors claim that the average memory access time can be reduce by the proposed PE, please provide more detail data by different aspect ratios.
  4. Figure 20 shows the completed RAM design, can author provide the detail performance of this RAM, and please discuss the role of PE in detail.
  5. How does the performance of this PE change under different processes? Are there any design considerations of PE under different processes?
  6. I suggest that author should provide the performance comparison with the other PE design in the published journal/conference.

Author Response

Dear the reviewer, 

The authors wish to thank you for providing valuable comments and suggestions. Every comment or suggestion has been responded in a revised version of the manuscript. Please see the attachment. 

Best regards,
Toru Tanzawa 

Reviewer 2 Report

This is an interesting paper dealing with pulse design techniques for memories. However, there are questions that make me reject it for publication.

-The figure captions can be better explained in general. No details are given.

- The quality of some figures is really bad. The authors should use vector graphics to solve this problem.

-No technological details are given of the measured and characterized technology. What is the metal employed? How is the fabrication process?

-At the beginning of the paper it seems that RRAM are going to be evaluated in one way or another. At the end, it seems that Figure 20 reflects an evaluation of a RAM module. The writting in general is confusing.

-No units are given thought the paper.

Author Response

(The authors gave the same response as above.)
